# Enhanced Stability and Improved Oral Absorption of Enzalutamide with Self-Nanoemulsifying Drug Delivery System

**DOI:** 10.3390/ijms25021197

**Published:** 2024-01-18

**Authors:** Su-Min Lee, Jeong-Gyun Lee, Tae-Han Yun, Jung-Hyun Cho, Kyeong-Soo Kim

**Affiliations:** 1Department of Pharmaceutical Engineering, Gyeongsang National University, 33 Dongjin-ro, Jinju 52725, Republic of Korea; m8121@naver.com (S.-M.L.); leepipi87@naver.com (J.-G.L.); xogks7702@naver.com (T.-H.Y.); 2Department of Pharmaceutical Engineering, Dankook University, 119 Dandae-ro, Dongnam-gu, Cheonan 31116, Republic of Korea

**Keywords:** enzalutamide, self-nanoemulsifying drug delivery system, nanoemulsion, solubility, dissolution, oral absorption

## Abstract

The purpose of this study is to develop and evaluate a self-nanoemulsifying drug delivery system (SNEDDS) to improve the oral absorption of poorly water-soluble enzalutamide (ENZ). Considering the rapid recrystallization of the drug, based on solubility and crystallization tests in various oils, surfactants and co-surfactants, Labrafac PG 10%, Solutol HS15 80%, and Transcutol P 10%, which showed the most stable particle size and polydispersity index (PDI) without drug precipitation, were selected as the optimal SNEDDS formulation. The optimized SNEDDS formulation showed excellent dissolution profiles for all the drugs released at 10 min of dissolution due to the increased surface area with a small particle size of approximately 16 nm. Additionally, it was confirmed to be stable without significant differences in physical and chemical properties for 6 months under accelerated conditions (40 ± 2 °C, 75 ± 5% RH) and stressed conditions (60 ± 2 °C). Associated with the high dissolutions of ENZ, pharmacokinetic parameters were also greatly improved. Specifically, the AUC was 1.9 times higher and the C_max_ was 1.8 times higher than those of commercial products (Xtandi^®^ soft capsule), resulting in improved oral absorption. Taken together with the results mentioned above, the SNEDDS could be an effective tool as a formulation for ENZ and other similar drugs.

## 1. Introduction

Enzalutamide (ENZ) is a prostate cancer treatment that inhibits the growth of prostate cancer and induces death by competitively inhibiting the binding of androgens to the receptor [1,2,3]. Despite these therapeutic benefits, ENZ is a class II molecule per the Biopharmaceutics Classification System due to its poor water solubility and high permeability [4,5,6]. Moreover, with a hydrophobic and quickly crystallizing character, this drug has an extremely irregular and very low dissolution profile in gastrointestinal (GI) fluids, resulting in highly unpredictable bioavailability [7]. To improve the solubility and oral absorption rate of poorly soluble drugs such as ENZ, numerous efforts have been made to increase its bioavailability by using various techniques, including solid dispersion, the use of a cocrystal, the use of an inclusion complex, salt formation, nanomilling, particle size reduction, and lipid-based formulation [8,9,10]. Among these, lipid-based drug delivery systems have recently attracted attention for improving drug solubility and bioavailability. The lipid-based formulation is pre-dissolving drugs in lipid excipients, so it can effectively avoid the dissolution step and a potentially rate-limiting dissolution step in the GI tract, eventually having the advantage of improving bioavailability [11,12]. The solid lipid nano-carrier, liposome, nano-structured lipid carrier, emulsion, and self-nanoemulsifying drug delivery system (SNEDDS) represent this lipid-based formulation [13]. SNEDDS is a homogeneous mixture of oil, surfactant, and cosurfactant. It can form an emulsion in an aqueous phase such as in the gastrointestinal tract even with light agitation and has a particle size in the range of ≤200 nm when dispersed (Figure 1) [14,15,16].

In this research, we aimed to improve the solubility of ENZ by applying SNEDDS technology. Although the issue of recrystallization of the drug in aqueous solution cannot be completely resolved, a composition that can suppress the rapid recrystallization of ENZ as much as possible was selected through solubility and crystallization tests. For the selected optimal SNEDDS formulation, physicochemical properties such as particle size, PDI, solubility, and dissolution rate were confirmed, and finally, improvement in bioavailability was evaluated through in vivo rat PK.

## 2. Results

### 2.1. Solubility and Crystallization Test

The SNEDDS formulation comprises oil, surfactant, and co-surfactant, each contributing to enhanced drug solubility, facilitated emulsion formation, and stabilized emulsions, respectively. Thus, a solubility test for ENZ was performed to select a suitable oil, surfactant, and co-surfactant for the composition of SNEDDS in Figure 2. Additionally, given the rapid recrystallization properties of ENZ, the crystal formation inhibition ability of the excipient for the drug was further evaluated, as shown in Figure 3. For the solubility test, 10 mg of ENZ was added per 1 mL of each excipient, while for the crystallization test, 160 mg of ENZ dissolved in acetone was added per 1 g of each excipient, and the initial sample was completely cleared. Among the oils, the solubility of ENZ in Labrafac PG was relatively high, at 3027.86 ± 70.03 μg/mL (Figure 2A). Other oils showed similar solubility at about 1000 μg/mL. In the oil crystallization test, ENZ crystals were observed to form in all tested oils (Figure 3A).

SNEDDS can enhance emulsion stability via the use of a high-HLB-value surfactant with a strong affinity for the hydrophilic portion and a low-HLB-value surfactant with a strong affinity for the hydrophobic portion [17]. Thus, surfactants were categorized in this study based on their HLB values, with those having an HLB value ≥ 10 being classified as surfactants (TPGS, Tween 80, Labrasol, Cremophor EL, RH60 and Solutol HS15), and those with an HLB value ≤ 10 being classified as co-surfactants (Span 80, Capryol 90, Transcutol P and Lauroglycol 90). In the surfactant solubility test, all surfactants, except TPGS, exhibited similar solubility values of around 500 µg/mL (Figure 2B), and among them, only Solutol HS15 inhibited ENZ crystal formation (Figure 3B). Among the co-surfactants, Transcutol P exhibited the most significant inhibition of crystal formation.

In the SNEDDS composition, oil is added for the purpose of increasing the solubility of the lipophilic drug. Therefore, an oil was selected based on the solubility evaluation results. However, recrystallization of the drug cannot be completely resolved when it is exposed to an aqueous solution. Therefore, a surfactant and co-surfactant that showed minimal crystal formation were selected to suppress the rapid recrystallization of drugs dissolved in oil. Based on the results described above, Labrafac PG as an oil, Solutol HS 15 as a surfactant, and Transcutol P as a co-surfactant were selected for further studies.

### 2.2. Characterization of SNEDDS

The primary focus in the manufacturing of an ENZ-loaded SNEDDS is to inhibit the recrystallization of ENZ. Based on the selected composition, a Labrafac PG and Solutol HS15 mixture with ENZ (40 mg equivalent dissolved in acetone) was characterized to select the appropriate weigh ratio for recrystallization inhibition. Labrafac PG and Solutol HS15 were mixed at a weight ratio of 1/9~9/1, and the particle size of dispersion was measured. In most mixtures, as the weight ratio of Solutol HS15 increased, miscibility was good, drug crystals were not formed and the particle size tended to decrease. Labrafac PG/Solutol HS15 at a weight ratio of 1/9 to 4/6 showed relatively small particle sizes of <50 nm (Figure 4). Therefore, the final SNEDDS ratio was selected by adding Transcutol P to the higher surfactant ratio of 1/9 and 2/8. As a result, as Labrafac PG and Transcutol P decreased, the dispersion in distilled water (D.W) was clear, and the particle size tended to decrease at about 20 nm (Table 1, Figure 5). Thus, Labrafac PG, Solutol HS15 and Transcutol P of 1/8/1 (ENZ8), with the lowest oil and co-surfactant content, were selected to form the SNEDDS composition with suppressed recrystallization.

Additionally, representative TEM images of ENZ8 dispersed in pH 6.8 buffer with 1% polysorbate 80 are presented in Figure 6. Representative single spherical and translucent nanoemulsion droplets measuring < 20 nm were observed in ENZ8. This value is similar to that when particle size was analyzed using a dynamic light scattering device. The small particle size of ENZ8 with no ENZ precipitation may have contributed to the high dissolution rate [18].

### 2.3. Dissolution Tests

The in vitro dissolution profiles of ENZ from the SNEDDS formulation were monitored and compared with those of the ENZ powder and commercial product (Xtandi^®^ soft capsule) under different pH conditions, pH 1.2, pH 4.0, and pH 6.8, and 0.3% cetyltrimethylammonium bromide in 0.1N HCl (Xtandi^®^ soft capsule FDA conditions) for 2 h. Under pH 1.2, 4.0, and 6.8 conditions, the dissolutions of ENZ powder were barely detected due to its low solubility, and the commercial product showed a low dissolution rate of about 31% or less (Figure 7). On the other hand, in the pH 1.2 solution, that of ENZ8 formulated into the liquid SNEDDS was 96.22 ± 3.72% within 10 min, showing a very rapid dissolution profile, followed by a final dissolution of 73.67 ± 0.58% at 120 min. There was a tendency for dissolution to decrease with recrystallization after 45 minutes, and recrystallization of ENZ could be confirmed through the PXRD pattern (Appendix A). However, compared to that in other studies related to ENZ, this does not appear to be a drastic decrease [19,20]. In addition, even though the dissolution rate is reduced, it shows a solubility that is about four times higher than that of commercial products. Similar results to those at a pH of 1.2 were also obtained at a pH of 4.0 and a pH of 6.8. At a pH of 4.0 and a pH of 6.8, the dissolution values of ENZ in the SNEDDS system were 95.49 ± 0.76% and 95.40 ± 3.96%, respectively, for the initial 10 min. In contrast, that of the Xtandi^®^ soft capsule was very low at 24.35 ± 0.14% and 16.57 ± 0.03%, respectively. According to the solubility results (Appendix A), the solubility of ENZ is independent of the pH conditions. Therefore, it is estimated that the dissolution profiles at a pH of 1.2, a pH of 4.0, and a pH of 6.8 are all similar. The dissolutions of ENZ8 at the initial 10 min improved 9.30 times (pH 1.2), 3.92 times (pH 4.0) and 5.76 times (pH 6.8) compared to those of the Xtandi^®^ soft capsule, and the final dissolution rates (at 120 min) improved 4.06 times (pH 1.2), 2.86 times (pH 4.0), and 4.58 times (pH 6.8). The improved dissolution rates of the drugs contained in SNEDDS formulations were reported in several documents. These results showed that the improved dissolutions of the SNEDDS formulation containing ENZ were because of the increased specific surface area of nano-sized emulsion droplets and the solubilization effects of optimal oils and surfactants [21,22]. As such, the SNEDDS formulation showed an improved dissolution profile compared to that of the commercial product, which is expected to improve the oral absorption of ENZ.

### 2.4. Stability Test

To evaluate the stability of the optimized SNEDDS formulation (ENZ8), the physico-chemical properties were checked while storing the samples under the accelerated stability test conditions (40 ± 2 °C, 75 ± 5% RH) and stressed stability test conditions (60 ± 2 °C) (Table 2). During the storage period, the formulation’s physical stability was confirmed to have maintained its initial state. For 6 months, the appearance of the emulsion remained clear without drug precipitation and phase separation (Figure 8A). The particle size slightly increased from 15.92 ± 0.13 to 19.05 ± 4.75 nm, as did the PDI from 0.03 ± 0.01 to 0.19 ± 0.03, but the nano-sized dispersion still remained stable (Table 2). The drug contents were 100.8 ± 0.9% and 100.2 ± 0.5% under accelerated storage and stressed storage, respectively, which are not significantly different from the initial value (101.2 ± 1.0%) (Figure 8B). In addition, the dissolution rate was shown to be 98.06 ± 0.74% and 98.67 ± 1.51% under accelerated storage and stressed storage, respectively, with no significant change from the initial dissolution rate (98.77 ± 4.24%) (Figure 8C). Because the SNEDDS system does not contain water in the pharmaceutical formulation, it is generally known to have relatively high stability compared to that of other liquid type dosage formulations [23,24] Taken together, these results show that the self-nanoemulsification behavior was maintained and that the physical properties of the SNEDDS formulation did not change significantly.

### 2.5. In Vivo Pharmacokinetic Study

In this study, the pharmacokinetic profile of an ENZ-loaded SNEDDS formulation (ENZ8) was compared to that of free ENZ and a commercial product (Xtandi^®^ soft capsule) in rats. The graph in Figure 9 shows the mean plasma concentration of ENZ versus time for free ENZ, the Xtandi^®^ soft capsule, and ENZ8 when given orally at a dose of 50 mg/kg of ENZ. The mean pharmacokinetic parameters (C_max_, T_max_, AUC, K_el_, and t_1/2_) of ENZ absorption are summarized in Table 3. The results showed that the ENZ8 formulation resulted in significantly higher plasma concentrations compared to those of the free drug and Xtandi^®^ soft capsule (*p* < 0.05). Specifically, the C_max_ (9.5 ± 2.3 μg/mL) and AUC (291.5 ± 43.4 h × μg/mL) values of the ENZ8-treated group were 1.8-fold and 1.9-fold higher than those of the Xtandi^®^ soft capsule-treated group, respectively. The observed increase in the oral bioavailability of ENZ in the SNEDDS could be attributed to a noticeable improvement in the dissolution rate and the subsequent absorption of the drug in rats.

## 3. Materials and Methods

### 3.1. Materials

ENZ was purchased from MSN Labratories Private Ltd. (Telangana, India). The Xtandi^®^ soft capsule, TPGS and MCT oil were supplied by Boryung Corporation (Seoul, Republic of Korea). Labrafac PG, Peceol, Labrasol, Capryol 90, Transcutol P, and Lauroglycol 90 were obtained from Gattefosse (St. Priest, France). Cremophor EL, Cremophor RH60, and Solutol HS15 were supplied by BASF (Ludwigshafen, Germany). Castor oil, coconut oil, olive oil, sesame oil, sunflower oil, corn oil, soybean oil, cotton seed oil, peanut oil, polysorbate 80, ammonium acetate, acetone, and acetonitrile were purchased from Daejung Co. Ltd. (Siheung, Republic of Korea). Sodium carboxymethyl cellulose was purchased from Duksan Pure Chemical (Ansan, Republic of Korea). All other chemicals used were of analytical grade.

### 3.2. HPLC Conditions

The HPLC analysis of samples was conducted using an Agilent 1260 Infinity HPLC system (Agilent Technologies, Santa Clara, CA, USA) equipped with a UV–Vis detector (Agilent G1314 1260, Agilent Technologies, Santa Clara, CA, USA). ENZ was separated using a reversed-phase column (VDSpher 100 C18 M-E column, 5 μm, 4.6 × 150 mm). The mobile phase was composed of 20 mM ammonium acetate aqueous solution and acetonitrile adjusted to pH 4.6 using acetic acid (45/55, *v*/*v*). HPLC analysis was performed with a flow rate of 1.3 mL/min. The injection volume was 10 μL, and UV detection was monitored at 235 nm [25]. Data acquisition and processing were carried out using OpenLab CDS CS C.01.08 Chemstation software.

### 3.3. Solubility Test

The solubility of ENZ in various excipients such as oils, surfactants, and co-surfac-tants was measured. Briefly, 10 mg of drug was added to 1 mL of each pure oil or 10% (*w*/*v*) surfactant solution [26,27,28]. The mixture containing the drug was mixed using a vortex and stored in a shaking water bath for 5 days at 37 °C and 50 rpm. The sample was centrifuged at 13,500 rpm for 10 min to remove the amount of undissolved drug. The supernatant solution was then obtained from the mixture, passed through a 0.45 µm membrane filter and diluted with acetonitrile [26,27,28]. The amount of ENZ was analyzed using the HPLC conditions described in Section 3.2. The experiment was repeated in triplicate.

### 3.4. Crystallization Test

ENZ nucleation was induced in various oils, surfactants, and co-surfactants to identify a compatible excipient for inhibiting the crystallization of ENZ. First, A solution was prepared by dissolving 160 mg of ENZ in 1 mL of acetone. Briefly, 3 mL of the solution was added to 3 g of each oil, surfactant, and co-surfactant. The sample containing the drug was stirred and stored at 60 °C for 1 day. Then, crystal formation and quantity in each sample were visually compared.

### 3.5. Preparation of SNEDDS

Considering the solubility results for ENZ in the solubility test, Labrafac PG, Solutol HS15, and Transcutol P were selected as the oil, surfactant and co-surfactant, respectively. First, in order to select the ratio of specific oils and surfactants that exhibited the acceptable dispersed particle size, various mixture solutions of surfactant and oil containing ENZ (40 mg equivalent dissolved in acetone) were prepared. Oil and surfactant mixtures were combined at various weight ratios (1/9~9/1), and mixed by stirring in 50 °C. The particle size of the dispersion was measured using the method described in Section 3.6. The oil content was minimized (10% and 20% of the total composition) based on the particle size of the drug in the oil and surfactant mixture. Then, the final SNEDDS ratio was selected by adding a co-surfactant. The particle size of the dispersion was measured using the method described in Section 3.6.

### 3.6. Particle Size Analysis

An amount of 200 μL of the prepared nanoemulsions was added to 300 mL of distilled water [27]. The particle size was determined using a particle size analyzer (Malvern Korea, Malvern, UK) at a wavelength of 633 nm and a scattering angle of 173°. The temperature was set to 37 °C. The hydrodynamic diameter was measured in triplicate.

### 3.7. Transmission Electron Microscopy (TEM) Analysis

The representative morphology and particle size of dispersed particles of ENZ in pH 6.8 buffer with 1% polysorbate 80 were analyzed using a transmission electron microscope (Talos L120C, Thermo Fisher Scientific Inc., Waltham, MA, USA). The ENZ SNEDDS diluted 5-fold in pH 6.8 buffer with 1% polysorbate 80 was dispensed dropwise onto a carbon-coated TEM grid (FCF200-CU-50, Electron Microscopy Sciences, Inc., Hatfield, PA, USA) and dried for 30 min in a desiccator [28]. Then, the prepared sample was scanned at the accelerating voltage of 300 kV.

### 3.8. Dissolution Test

The dissolution test of ENZ powder, the Xtandi^®^ soft capsule solution, and the optimized SNEDDS formulation (ENZ8) was performed using an RC-8D dissolution testing apparatus (Dream Test, Seoul, Republic of Korea). Each formulation containing 40 mg of ENZ was placed into the dissolution tester. The dissolution test was performed under conditions of a 50 rpm paddle speed in 900 mL of pH 1.2 medium, pH 4.0 medium, pH 6.8 medium, or 0.1N HCl with 0.3% cetyltrimethylammonium bromide at 37.0 ± 0.5 °C. The samples were aliquoted in 3 mL increments with a syringe and collected at predetermined time intervals (5, 10, 15, 30, 45, 60, 90, and 120 min). The aliquoted sample was diluted 2-fold with acetonitrile and filtered through a 0.2 μm syringe filter. The ENZ amount in the filtered sample was analyzed using the HPLC conditions described in Section 3.2.

### 3.9. Stability Test

To investigate the stability of ENZ in the optimized SNEDDS formulation (ENZ8), ENZ8 was packed into sample vials and placed in a chamber with a constant temperature and humidity until sampling and analysis. Accelerated and stressed stability tests were conducted for ENZ8 over a period of 6 months. For the accelerated stability test, temperature and relative humidity were maintained at 40 ± 2 °C and 75 ± 5%, respectively; for the stressed stability test, temperature was maintained at 60 °C. Physical appearance factors such as phase separation, drug precipitation, particle size, PDI value, dissolution profile, and drug content were assessed [29].

### 3.10. In Vivo Pharmacokinetic Study

The in vivo pharmacokinetics of the optimized SNEDDS formulation (ENZ8) were assessed in female Sprague–Dawley rats (age: 9–10 weeks old; weight: 250 ± 20 g) purchased from Samtaco (Osan, Republic of Korea). All animals were acclimatized to a standard laboratory environment (25 average temperature and 12/12 h light/dark cycle) for one week before the experiments and given access to food and water ad libitum. The experimental protocols were reviewed and approved by the Institutional Animal Care and Use Committee of Gyeongsang National University (approval no. GNU-230719-R0149; approval date: 19 July 2023).

Rats were anesthetized using CO_2_ gas, followed by cannulation of the femoral artery using polyethylene tubes (Natsume Seisakusho, Tokyo, Japan) for blood sampling. Rats were orally administered the ENZ dispersion (50 mg/mL) in 0.5% carboxymethyl cellulose or a freshly prepared Xtandi^®^ soft capsule and liquid ENZ8 at a dose of 50 mg/kg. Blood samples (350 μL) were withdrawn from the femoral artery at 0.25, 0.5, 1, 2, 3, 6, 9, and 12 h and 1, 2, and 3 days and immediately centrifuged at 13,500 rpm for 15 min at 4 °C [30]. Finally, plasma in the supernatant was separated and stored at −20 °C until analysis.

Pharmacokinetic plasma samples were quantified for ENZ concentration using the HPLC analysis described in Section 3.2, with an injection volume of 40 μL. Briefly, 150 μL of the plasma sample was mixed with 50 μL of the internal standard solution (20 μg/mL of tadalafil in acetonitrile) and 300 μL of acetonitrile [31]. The sample was vortexed for 1 min to achieve deproteinization and drug extraction. The samples were then centrifuged at 13,500 rpm for 15 min at 4 °C; the supernatant was filtered using a 0.2 μm filter and transferred to an analytical vial for HPLC analysis. Similarly, a series of calibration standards in rat plasma were prepared by combining 150 μL of blank rat plasma with 50 μL of standard solution in acetonitrile. ENZ concentrations of 0.01, 0.05, 0.1, 0.5, 1, 5, 10, and 50 μg/mL and 250 μL of acetonitrile were added, followed by processing similar to that mentioned for the plasma samples above. Non-compartmental analysis was used to calculate various pharmacokinetic parameters. These included the area under the plasma concentration versus time curve from zero to 72 h (AUC 0→72 h), the maximum plasma concentration (C_max_), the time required to reach C_max_ (T_max_), the elimination rate (K_el_), and the half-life (t_1/2_).

## 4. Conclusions

ENZ is a fast-crystallizing, hydrophobic compound that has solubility-limited absorption in vivo. The optimal SNEDDS composition that can suppress rapid recrystallization characteristics was developed using Labrafac PG, Solutol HS15, and Transcutol P. This formulation formed a stable nanomulsion system for at least 6 months under accelerated storage conditions (40 ± 2 °C, 60 ± 5% RH) and stressed storage conditions (60 ± 2 °C). In addition, it showed excellent dissolution profiles in all pH solutions. The in vivo pharmacokinetic results also showed higher oral absorption of ENZ than that of ENZ powder and even the commercial product. In conclusion, this system suggests the potential use of the SNEDDS for the oral administration of ENZ or other hydrophobic drugs.

## Figures and Tables

**Figure 1 ijms-25-01197-f001:**
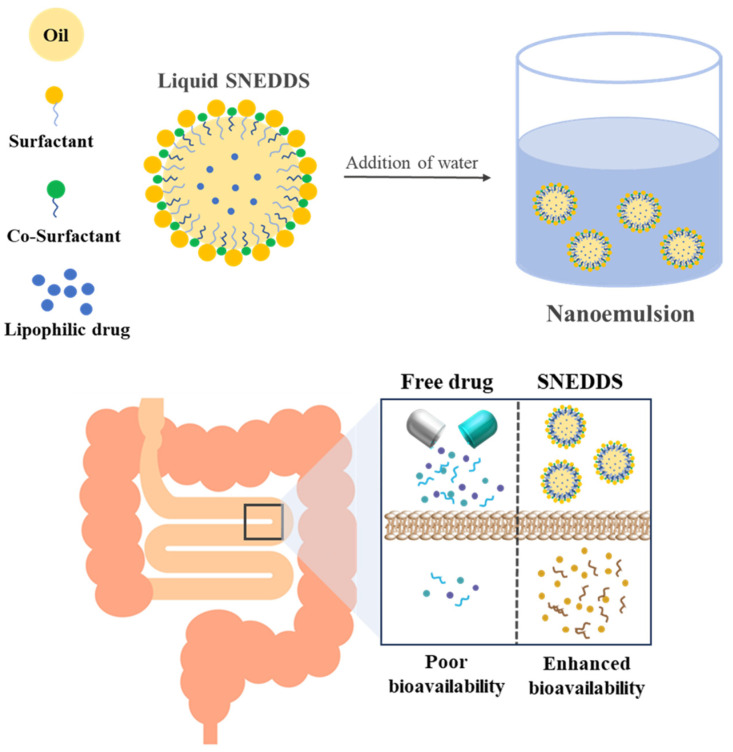
A diagram illustrating the structural characteristics of SNEDDS and the form of SNEDDS upon dispersion in water.

**Figure 2 ijms-25-01197-f002:**
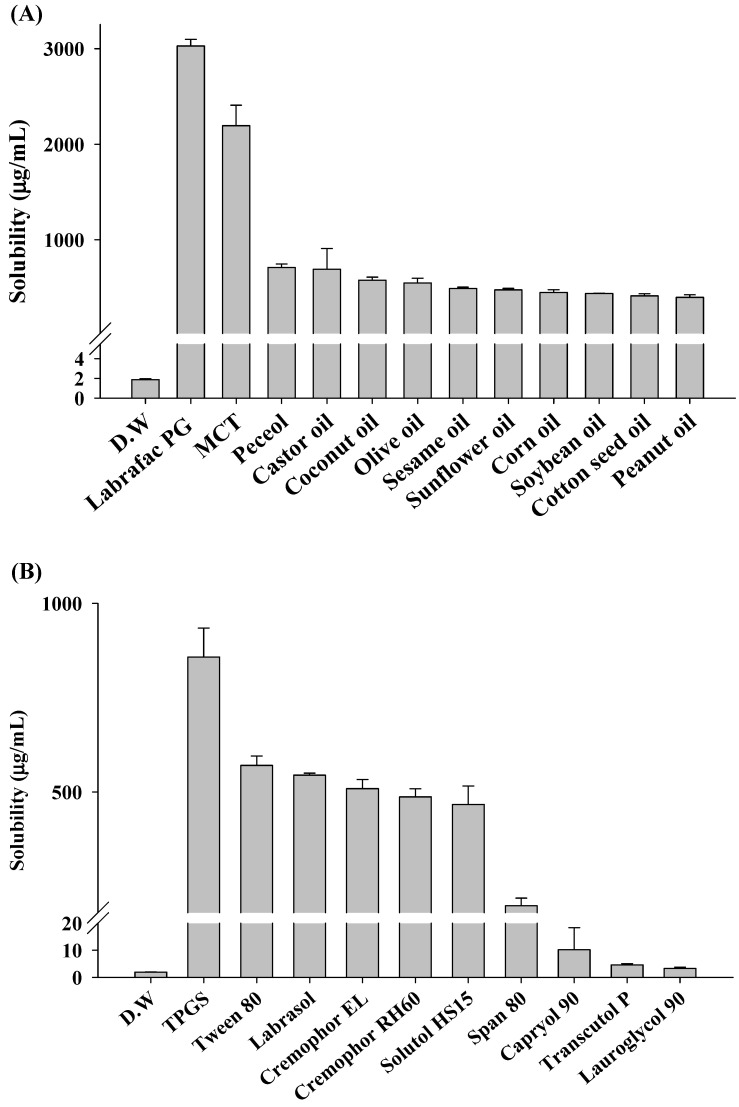
The solubility of ENZ in various (**A**) oils and (**B**) 10% (*w*/*v*) surfactants.

**Figure 3 ijms-25-01197-f003:**
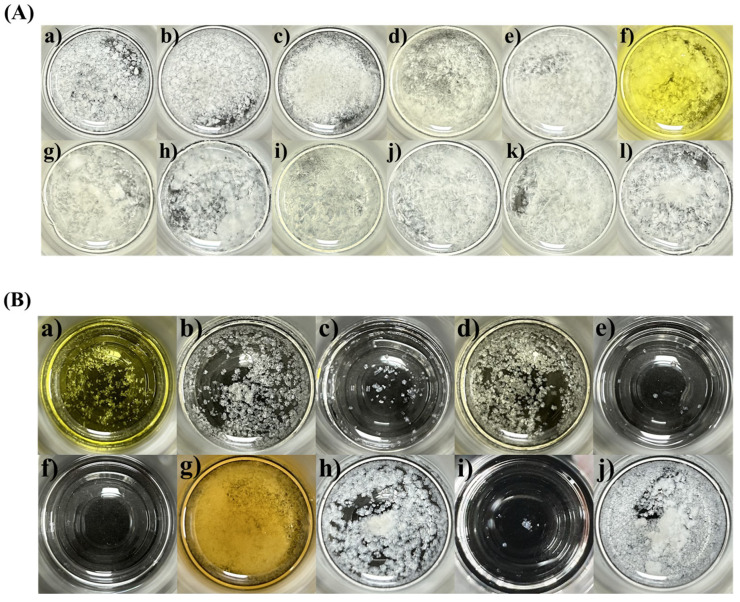
Image of ENZ crystals in various (**A**) oils and (**B**) surfactants. (**A**): (**a**) Labrafac PG; (**b**) MCT; (**c**) Peceol; (**d**) castor; (**e**) coconut; (**f**) olive; (**g**) sesame; (**h**) sunflower; (**i**) corn; (**j**) soybean; (**k**) cotton seed; (**l**) peanut. (**B**): (**a**) TPGS; (**b**) Tween 80; (**c**) Labrasol; (**d**) Cremophor EL; (**e**) Cremophor RH60; (**f**) Solutol HS15; (**g**) Span 80; (**h**) Capryol 90; (**i**) Transcutol P; (**j**) Lauroglycol 90.

**Figure 4 ijms-25-01197-f004:**
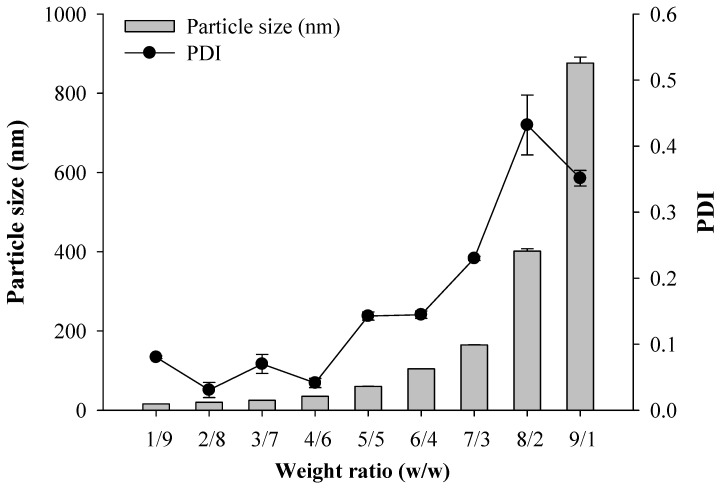
The ENZ particle size and PDI value of water dispersion, for the oil–surfactant mixture.

**Figure 5 ijms-25-01197-f005:**
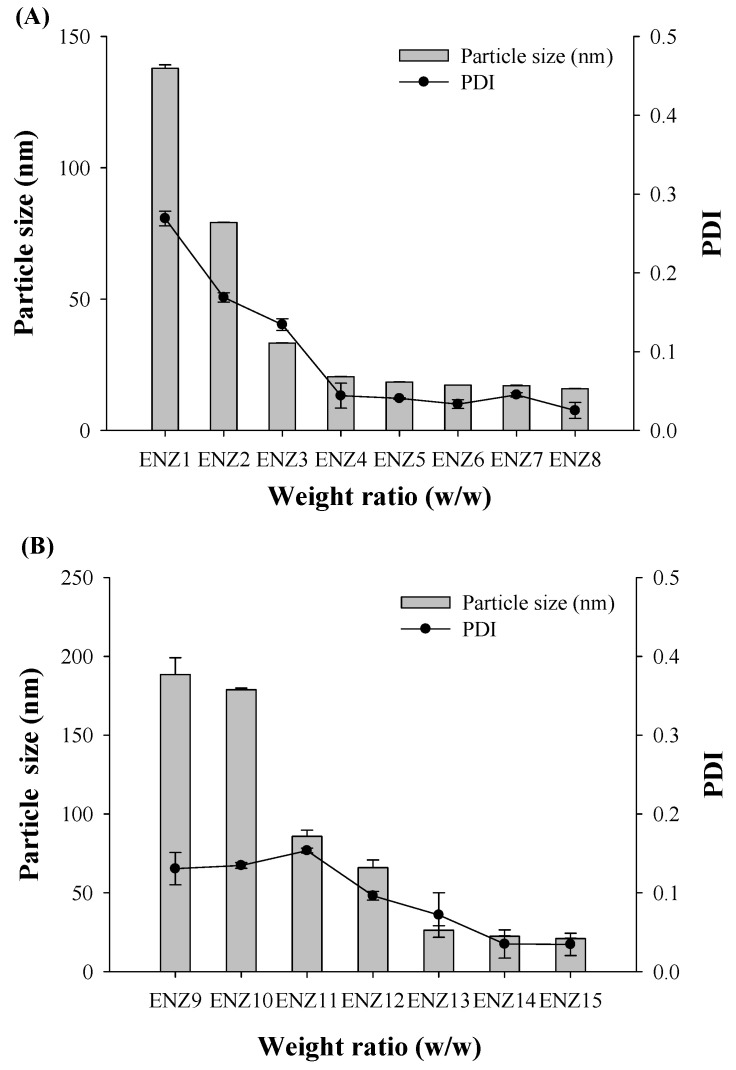
The ENZ particle size and PDI for the SNEDDS composition with (**A**) 10% oil content and (**B**) 20% oil content.

**Figure 6 ijms-25-01197-f006:**
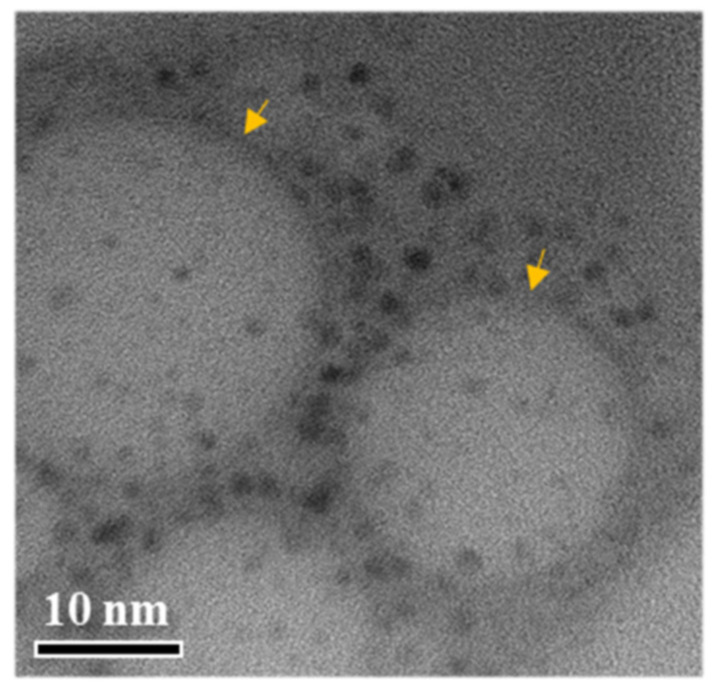
Transmission electron microscopy image of ENZ SNEDDS (ENZ8).

**Figure 7 ijms-25-01197-f007:**
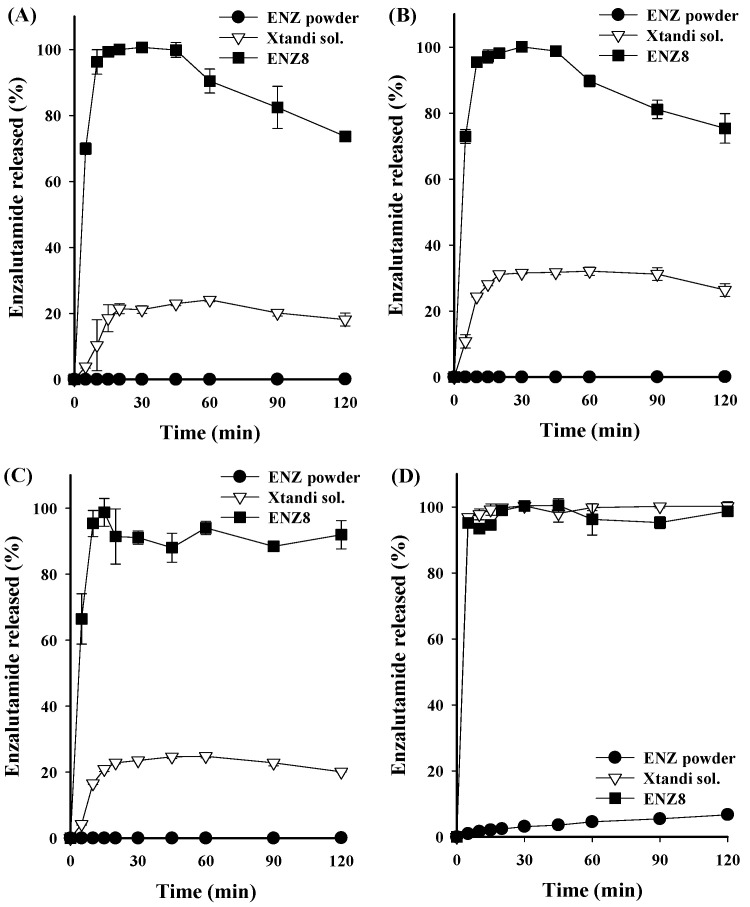
In vitro dissolution profiles of ENZ powder, Xtandi^®^ soft capsule, and ENZ8. Dissolution profile under (**A**) pH 1.2, (**B**) pH 4.0, (**C**) pH 6.8, and (**D**) 0.1N HCl with 0.3% cetyltrimethylammonium bromide.

**Figure 8 ijms-25-01197-f008:**
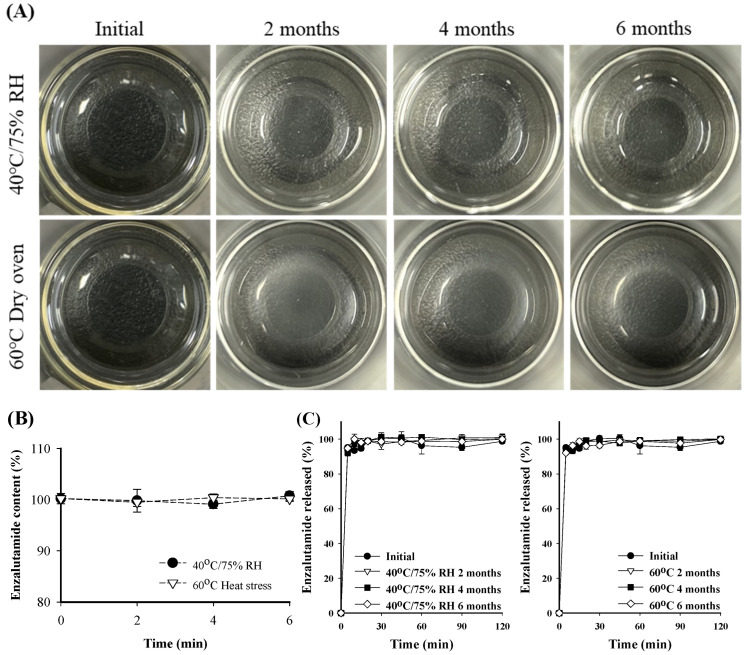
The ENZ8 (**A**) appearance, (**B**) drug content and (**C**) dissolution profiles in 0.1N HCl with 0.3% cetyltrimethylammonium bromide over 6 months under accelerated (40 ± 2 °C, 75 ± 5% RH) and heat-stressed (60 ± 2 °C) conditions.

**Figure 9 ijms-25-01197-f009:**
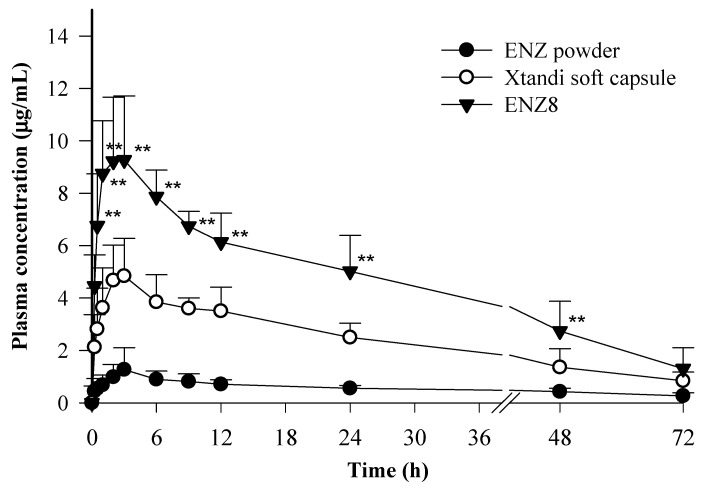
Plasma concentration–time profiles in rats after oral administration of ENZ powder, Xtandi^®^ soft capsule and optimized SNEDDS formulation (ENZ8); ** *p* < 0.05 compared with Xtandi^®^ soft capsule.

**Table 1 ijms-25-01197-t001:** The ENZ particle size and PDI value of water dispersion, for the SNEDDS compositions.

Formulation No.	Vehicle Composition (Weight Ratio %)			Particle Size (nm)	PDI
	Labrafac PG	Solutol HS15	Transcutol P		
ENZ1	10	10	80	137.87 ± 1.30	0.27 ± 0.01
ENZ2	10	20	70	79.22 ± 0.15	0.17 ± 0.01
ENZ3	10	30	60	33.29 ± 0.11	0.13 ± 0.01
ENZ4	10	40	50	20.51 ± 0.09	0.04 ± 0.02
ENZ5	10	50	40	18.45 ± 0.05	0.04 ± 0.00
ENZ6	10	60	30	17.30 ± 0.00	0.03 ± 0.01
ENZ7	10	70	20	17.10 ± 0.13	0.05 ± 0.00
ENZ8	10	80	10	15.92 ± 0.13	0.03 ± 0.01
ENZ9	20	10	70	188.50 ± 10.66	0.13 ± 0.02
ENZ10	20	20	60	178.87 ± 1.01	0.13 ± 0.00
ENZ11	20	30	50	85.87 ± 3.92	0.15 ± 0.00
ENZ12	20	40	40	65.95 ± 4.89	0.10 ± 0.01
ENZ13	20	50	30	26.25 ± 2.87	0.07 ± 0.03
ENZ14	20	60	20	22.47 ± 0.13	0.04 ± 0.02
ENZ15	20	70	10	20.99 ±0.12	0.03 ± 0.01

**Table 2 ijms-25-01197-t002:** Stability test of SNEDDS at 40 ± 2 °C, 75 ± 5% RH, and 60 ± 2 °C (mean ± S.D.; *n* = 3).

Time	Stability Assessment			
	Particle Size (nm)	PDI	Dissolution Rate (%)	Drug Content (%)
Initial	15.92 ± 0.13	0.03 ± 0.01	98.77 ± 4.24	101.2 ± 1.0
40 °C/75% RH 2 months	17.68 ± 0.10	0.18 ± 0.01	97.46 ± 3.06	100.9 ± 3.3
40 °C/75% RH 4 months	17.93 ± 0.62	0.19 ± 0.03	97.76 ± 1.59	101.1 ± 0.8
40 °C/75% RH 6 months	19.05 ± 4.75	0.14 ± 0.07	98.06 ± 0.74	100.8 ± 0.9
60 °C Heat stress 2 months	16.15 ± 0.34	0.03 ± 0.01	98.00 ± 1.41	102.8 ± 0.8
60 °C Heat stress 4 months	16.12 ± 0.16	0.07 ± 0.04	97.73 ± 0.53	100.4 ± 0.7
60 °C Heat stress 6 months	18.66 ± 1.23	0.05 ± 0.01	98.67 ± 1.51	100.2 ± 0.5

**Table 3 ijms-25-01197-t003:** Pharmacokinetic parameters of ENZ powder, Xtandi^®^ soft capsule and ENZ8 at an equivalent dose of 50 mg/kg in rats (mean ± S.D.; *n* = 6).

PK Parameters	ENZ	Xtandi^®^ Soft Capsule	ENZ8 (SNEDDS)
AUC_0–72_ (μg∙h/mL)	37.5 ± 6.4	151.4 ± 36.5	291.5 ± 43.4
C_max_ (μg/mL)	1.4 ± 0.8	5.1 ± 1.2	9.5 ± 2.3
T_max_ (h)	3.8 ± 1.7	3.5 ± 2.8	2.1 ± 1.1
K_el_	0.02 ± 0.01	0.03 ± 0.01	0.03 ± 0.01
T_1/2_	47.3 ± 25.7	30.2 ± 8.5	27.7 ± 12.2

## Data Availability

Data available on request due to restrictions, e.g., privacy or ethical restrictions.

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
