# Peer review of "Enhanced Stability and Improved Oral Absorption of Enzalutamide with Self-Nanoemulsifying Drug Delivery System"

_ijms, 2024, doi:10.3390/ijms25021197_

Round 1

Reviewer 1 Report

Comments and Suggestions for Authors

In the crystallization study, the amount taken was mostly less than the solubility of drug in oil. How did nucleation and crystallization occur in most of the surfactants and oils?

In the case of surfactants, the authors mentioned that selection was based on the absence of crystals. However, in case oil there was crystallization in the selected oil. How it will be justified?

The authors mentioned that…The key focus in the manufacturing of SNEDDS is to inhibit the recrystallization of ENZ. How this was proved. After dissolution whether any precipitate was collected and characterized for crystallinity?

There was a significant decrease in the dissolution after 20 min. decreased. This may be due to recrystallization of the drug. The author should justify this with proper analytical tools.

For stability studies, the samples were preconcentrates or diluted with water/buffer. Please clarify?

In TEM image, there were multiple droplets visible inside the globule. What are those multi globules or vescicles?

Comments on the Quality of English Language

The english language is ok

Author Response

Thank you very much for taking the time to review this manuscript.

Reviewer 2 Report

Comments and Suggestions for Authors

This manuscript reports on a self-nanoemulsifying drug delivery system (SNEDDS) designed to improve the oral absorption of poorly water-soluble enzalutamide (ENZ). To this end, the authors investigate an optimal formulation for encapsulating ENZ that prevents recrystallization of ENZ in the nano-emulsion through solubility and crystallization tests. The optimized SNEDDS improved the ENZ release properties. Furthermore, the pharmacokinetic parameters of the drugs in an oral administration were improved by using the optimized formulation compared to ENZ powder and commercial capsule products. The topic is appropriate in this journal and this study will bring a useful suggestion for formulating nano-emulsions encapsulating crystalline hydrophobic drugs. The following are the comments of this reviewer that should be addressed in a revision.

1)    The role of oil, surfactant, and co-surfactant in the formulation of nano-emulsions should be explained.

2)    The structure of ENZ SNEDDS(ENZ8) in the transmission electron microscope image (Figure 5) looks like particles within particles. A schematic representation of the structure of nano-emulsions showing the location of oil, surfactant, co-surfactant, and drugs would help the readers understand the structure.

3)    What is the significance of setting the pH 1.2, 4, and 6.8 in the dissolution test? Is release expected in these pH environments for oral absorption? Please explain.

4)    In Figure 6, it is unclear how the hydrophobic ENZ is rapidly released from the emulsions into the medium, even though the smaller emulsions have a larger specific surface area. Is the released drug dissolved in the aqueous medium or is it solubilized by the surfactants? In addition, it is unclear why the percentage of released ENZ decreased with time in the case of ENZ8. Please clarify and discuss these points.

5)    It is unclear why the dissolution test was performed in 0.3% cetyltrimethylammonium bromide in 0.1N HCl because there is no explanation for Figure 6 D). Please explain.

6)    Remove A) in Figure 6 C).

Comments on the Quality of English Language

Minor English editing is recommended.

Author Response

(The authors gave the same response as above.)
